# The Cost of Improving the Precision of the Variational Quantum Eigensolver for Quantum Chemistry

**DOI:** 10.3390/nano12020243

**Published:** 2022-01-14

**Authors:** Ivana Miháliková, Matej Pivoluska, Martin Plesch, Martin Friák, Daniel Nagaj, Mojmír Šob

**Affiliations:** 1Institute of Physics of Materials, Czech Academy of Sciences, Žižkova 22, CZ-616 62 Brno, Czech Republic; mihalikova@ipm.cz (I.M.); friak@ipm.cz (M.F.); mojmir@ipm.cz (M.Š.); 2Institute of Computer Science, Masaryk University, Šumavská 416, CZ-602 00 Brno, Czech Republic; plesch@savba.sk (M.P.); dnagaj2@gmail.com (D.N.); 3Department of Condensed Matter Physics, Faculty of Science, Masaryk University, Kotlářská 2, CZ-611 37 Brno, Czech Republic; 4Institute of Physics, Slovak Academy of Sciences, Dúbravská cesta 9, SK-841 04 Bratislava, Slovakia; 5Department of Chemistry, Faculty of Science, Masaryk University, Kotlářská 2, CZ-611 37 Brno, Czech Republic

**Keywords:** noisy quantum processors, variational quantum eigensolver, quantum chemistry

## Abstract

New approaches into computational quantum chemistry can be developed through the use of quantum computing. While universal, fault-tolerant quantum computers are still not available, and we want to utilize today’s noisy quantum processors. One of their flagship applications is the variational quantum eigensolver (VQE)—an algorithm for calculating the minimum energy of a physical Hamiltonian. In this study, we investigate how various types of errors affect the VQE and how to efficiently use the available resources to produce precise computational results. We utilize a simulator of a noisy quantum device, an exact statevector simulator, and physical quantum hardware to study the VQE algorithm for molecular hydrogen. We find that the optimal method of running the hybrid classical-quantum optimization is to: (i) allow some noise in intermediate energy evaluations, using fewer shots per step and fewer optimization iterations, but ensure a high final readout precision; (ii) emphasize efficient problem encoding and ansatz parametrization; and (iii) run all experiments within a short time-frame, avoiding parameter drift with time. Nevertheless, current publicly available quantum resources are still very noisy and scarce/expensive, and even when using them efficiently, it is quite difficult to perform trustworthy calculations of molecular energies.

## 1. Introduction

Computer simulations of quantum systems constitute a crucial tool for a deeper understanding of behavior and properties of matter at the atomic scale. However, when investigating the electronic structure of larger molecules, we quickly encounter the limits of classical computers. The space and time requirements for describing the states, and even more so for performing optimizations on them, grow prohibitively. Thus, we resort to many clever types of approximations [1,2,3,4,5,6,7,8,9]. Oftentimes, though, they are not good enough, or start to scale badly.

It is natural to imagine that using one quantum mechanical system to simulate another could be a more efficient approach. The concept of a quantum computer, based on the laws of quantum physics, was first proposed by Richard Feynman [10]. He envisioned that the way to deal with the exponential amount of information appropriate to study physical systems was to use quantum systems as computers themselves. We have come a long way since then, and today we have access to the first small, programmable quantum chips, together with 25 years of development of quantum algorithms [11,12,13] and protocols for simulation, optimization, and many other applications in sensing, cryptography and communication [14,15,16,17].

While fault-tolerant, universal quantum computing is still out of reach, much of today’s development focuses on demonstrating quantum supremacy in problems without direct applications [18,19,20]. Our goal here is less ambitious, but more practical, utilizing the imperfect computers we have at hand for a practical task. We investigate one of the key promised applications of quantum computing: the understanding of molecular structure. While the long-term goal is to find more efficient solutions for a problem deemed intractable on conventional computers for large problem sizes, we want to understand how well we can do today for small test cases.

The basic task is to solve the many-body Schrödinger equation
(1)$$\hat{H}\left|\psi \rangle =E\;\right|\psi \rangle ,$$
where $\hat{H}$ denotes the Hamiltonian operator, |ψ〉 is the system wave function and E represents the energy of the system. Finding the ground state energy of the studied system is a difficult optimization problem. However, we now know several promising approaches for applications in quantum chemistry [21], relying on natural quantum encodings of the problems, and algorithms utilizing superpositions and entanglement, resulting in efficient search and energy evaluation.

One way of approximately obtaining the minimum eigenvalue of a Hamiltonian is to start with an initial guess and iteratively search for optimal parameters. One such popular approach aimed at solving the many-particle problem is the hybrid classical-quantum variational quantum eigensolver (VQE) algorithm proposed by Peruzzo et al. [22]. It is an application of the time-independent variational principle that benefits from the cooperation of classical and quantum computers, and is suitable for use in near-term, imperfect quantum devices. For example, in [22], the authors used a small photonic quantum computer to calculate the He–H${}^{+}$ system’s ground-state energies for various atomic distances. Next, O’Malley et al. [23] studied the properties of the H${}_{2}$ molecule using Google’s digital quantum computer with superconducting qubits. The researchers used two different methods for finding the ground-state energies of H${}_{2}$: the phase-estimation algorithm (PEA) and the variational quantum eigensolver (VQE). The former method can in principle obtain the answer with arbitrary precision, but only if there are no errors in the process. As in practice, errors are always present, and so the VQE method works better. However, it also has its pitfalls, which we aim to elucidate.

In this work, we want to understand the possible sources of errors in VQE calculations when implemented on current, publicly available superconducting quantum processors. Our goal is to provide a recipe for the efficient use of the scarce resources, as well as practical guidance for the starting practitioner, with the aim to run quantum chemistry computations on current devices.

If our (superconducting) qubits were a closed system, the dynamical evolution of their state would be fully determined by their initial state and Hamiltonian. In reality, the system is partially open, the qubits interact with their environment, their internal interactions are not perfect, and so the overall state is not deterministic. A generalized theory of the interactions between a quantum state and its environment was derived by Bloch [24], manifesting as extra degrees of freedom we cannot control. These affect the state of the system and result in a loss of coherence [25]. Noise comes from controlling the qubits (e.g., via pulses applied to the qubits) or reading out the final states of the qubits (there is a finite probability that the recorded classical bit value will be flipped from the true outcome of a measurement). We also face errors from random fluctuations of parameters coupled to our qubits, such as (i) thermal voltage and current fluctuations in control lines or (ii) randomly fluctuating electric and magnetic fields in the local qubit environment.

Minimizing various sources of noise and errors is a very complex and demanding task involving materials science, fabrication engineering, electronics design, cryogenic engineering, and qubit design. Of course, our aim is fault-tolerant quantum processing, which is currently out of reach due to the precision and overhead requirements [26]. Meanwhile, we often employ methods for dealing with noisy data. There are many different approaches to correcting the noisy results (see Refs. [27,28,29,30,31,32,33,34,35,36] and references therein), based on different techniques of noise characterization.

Finally, even hypothetical noiseless quantum processors are prone to *stochastic noise*. Our calculations end with estimating the values of Pauli observables in the final state. This can only be achieved via sampling the outcome probabilities, with a limited number of shots. Moreover, stochastic noise comes in also through the heuristic nature of the classical optimizer part of the VQE algorithm. The good news is that we can investigate and control this noise, thanks to our selection of optimizer, and the parameters of the experimental runs. We will now attempt to characterize in detail how each source of noise influences the convergence of the VQE algorithm. We will then look at how to efficiently distribute our resources to achieve the desired precision of the calculations, if it is possible at all.

## 2. Methods

There are several preparatory steps before studying molecules on real quantum devices. We start with the molecular Hamiltonian in the second quantized form [37]. The fermionic Hamiltonian with one- and two-electron terms ($h_{ij}$ and $h_{ijkl}$) is given by
(2)$$\hat{H}=\sum_{ij}h_{ij}\;\;a_{i}^{†}a_{j}+\frac{1}{2}\sum_{ijkl}h_{ijkl}\;\;a_{i}^{†}a_{j}^{†}a_{l}a_{k},$$
where $a_{i}^{†}$ and $a_{j}$ are fermionic creation and annihilation operators. We then need to map this fermionic Hamiltonian into qubit operators represented in the Pauli operator basis [38]. This can be achieved in multiple ways, e.g., with the Bravyi–Kitaev [39] or Jordan–Wigner [40] transformations. Here, we chose the Bravyi–Kitaev transformation [39] of the hydrogen molecule Hamiltonian, produced using the publicly available qiskit package [41]. The code for this and all of our following calculations are located in [42]. We also invite the reader to read the introductory VQE tutorial [43].

Choosing the *STO-3G* basis set, and the distance between hydrogen atoms set to 0.725Å, not taking into account the Coulomb repulsiom between nuclei, we arrive at a 4-qubit Hamiltonian
(3)$$\begin{array}{cc} {\hat{H}}_{H_{2}} & =c_{0}\;1+c_{1}\;Z_{0}+c_{2}\;Z_{1}Z_{0}+c_{1}\;Z_{2}+c_{2}\;Z_{3}Z_{2}Z_{1}+c_{3}\;Z_{1}+c_{4}\;Z_{2}Z_{0}\\ \; & +c_{5}\;X_{2}Z_{1}X_{0}+c_{6}\;Z_{3}X_{2}X_{0}+c_{6}\;X_{2}X_{0}+c_{5}\;Z_{3}X_{2}Z_{1}X_{0}\\ \; & +c_{7}\;Z_{3}Z_{2}Z_{1}Z_{0}+c_{7}\;Z_{2}Z_{1}Z_{0}+c_{8}\;Z_{3}Z_{2}Z_{0}+c_{3}\;Z_{3}Z_{1}. \end{array}$$

The *Z* and *X* terms are Pauli operators, and the coefficients $c_{i}$ are integrals calculated using the qiskit.chemistry package [44]:(4)$$\begin{array}{ccc} c_{0}=-0.80718,\; & c_{1}=0.17374,\; & c_{2}=-0.23047,\\ c_{3}=0.12149,\; & c_{4}=0.16940,\; & c_{5}=-0.04509,\\ c_{6}=0.04509,\; & c_{7}=0.16658,\; & c_{8}=0.17511. \end{array}$$

Note that the Hamiltonian in Equation (Equation 4) commutes with $Z_{1}$ and with $Z_{3}$. Therefore, the Hamiltonian is block-diagonal with 4 blocks, each corresponding to a particular computational basis setting of qubits 1 and 3. This can be used to find a Hamiltonian with the same ground energy expressed in 2-qubit space [44]:(5)$${\hat{H}}_{H_{2}}=c_{0}\;1+c_{1}\;Z_{0}+c_{1}\;Z_{1}+c_{2}\;Z_{1}Z_{0}+c_{3}\;X_{1}X_{0},$$
with the coefficients
(6)$$\begin{array}{cc} c_{0}=-1.05016,\; & c_{1}=0.40421,\\ c_{2}=0.01135,\; & c_{3}=0.18038. \end{array}$$

With two equivalent formulations (both Hamiltonians have the same ground-state energy) of the problem in hand, we were able to investigate which form is more amenable to VQE optimization. They gave us a chance to explore various problem sizes, parametrized state ansatzes, energy landscapes, and investigate how noise affects the procedures, as the optimizations will involve different numbers of gates. Note that the choice of the STO-3G basis set here was deliberately chosen to keep the studied Hamiltonian relatively small. However, it is possible to formulate the qubit Hamiltonian of the hydrogen molecule using another, possibly larger, basis set. This naturally leads to a larger problem size; for example, consider the 6-31G basis set, which leads to 8-qubit Hamiltonian with 185 terms.

We could then run the VQE algorithm. Its input is a Hamiltonian expressed in terms of qubits and its aim is to find the eigenvector with the lowest eigenvalue. For this, the VQE iterates these four steps:Quantum: prepare a parametrized quantum state on a quantum device;Quantum: measure each Hamiltonian term (requires repetitions of step 1);Classical: sum the expectation values of the Hamiltonian terms to estimate the energy of the parametrized state;Classical: use the energy value to update the parameters of the trial quantum state.

until we met the convergence criteria of the classical optimization method.

To minimize the expectation value of the energy, we chose the simultaneous perturbation stochastic approximation (SPSA) [45,46,47], classical optimization method implemented in qiskit. SPSA is a pseudo-gradient method for optimizing problems with varying numbers of unknown parameters. SPSA starts with the initial vector of parameters ${\vec{\theta }}_{0}^{}$. In each iteration, the parameter vector is simultaneously shifted twice as
(7)$${\vec{\theta }}_{k}^{\pm }={\vec{\theta }}_{k}^{}\pm c_{k}^{}{\vec{\Delta }}_{k}^{},$$
where $c_{k}^{}$ is a preassigned positive number, k is the iteration number and ${\vec{\Delta }}_{k}^{}$ is a randomly generated vector (from the Bernoulli distribution). To approximate the gradient at ${\vec{\theta }}_{k}^{}$, we utilized the gradients at ${\vec{\theta }}_{k}^{+}$ and ${\vec{\theta }}_{k}^{-}$ as
(8)$${\vec{g^{}}}_{k}\left({\vec{\theta }}_{k}^{}\right)=\frac{\langle \psi ({\vec{\theta }}_{k}^{+})\left|H\right|\psi ({\vec{\theta }}_{k}^{+})\rangle -\langle \psi ({\vec{\theta }}_{k}^{-})\left|H\right|\psi ({\vec{\theta }}_{k}^{-})\rangle }{2c_{k}}{\vec{\Delta }}_{k}^{}.$$

In each iteration step, we thus need to measure the energies of two quantum states, prepared with parameter settings ${\vec{\theta }}_{k}^{+}$ and ${\vec{\theta }}_{k}^{-}$. We then updated the underlying parameters ${\vec{\theta }}_{k}^{}$ to
(9)$${\vec{\theta }}_{k+1}^{}={\vec{\theta }}_{k}^{}-a_{k}^{}{\vec{g}}_{k}^{}\left(\vec{\theta_{k}^{}}\right),$$
where $a_{k}^{}$ is a preassigned positive number and ${\vec{g}}_{k}^{}\left({\vec{\theta }}_{k}^{}\right)$ is the approximated gradient (Equation 8) that depends on ${\vec{\theta }}_{k}^{+}$ and ${\vec{\theta }}_{k}^{-}$.

The optimization procedure runs for a number of iterations controlled by the maxiter parameter (in qiskit’s implementation). We then updated the parameter vector ${\vec{\theta }}_{k}^{}$, and measured the final value of energy for the underlying optimized parameters. There is a subtlety that influences the total number of function evaluations. Qiskit’s implementation of SPSA includes additional initial exploration—a calibration phase that depends on the maximum number of iterations with $\min\left\{\mathrm{maxiter}/5,\;25\right\}$ steps.

There are many possible choices for the quantum circuit that prepares the trial state for simulating the molecules using VQE. We employed the hardware-inspired (easy to implement) $R_{y}R_{z}$ and $R_{y}$ variational forms, accompanied by linear entanglement [48]. The circuits consist of two main layers: (i) a layer of parametrized $R_{y}$ or $R_{y}$ and $R_{z}$ single-qubit rotations applied to each qubit in the quantum register, alternating with (ii) an entanglement-creating layer of CNOT gates. We could easily alternate these two layers, creating circuits of varying depths, increasing the complexity of the ansatz and the number of parameters to be optimized.

The simplest variational circuit of depth d=1 consists of just the entanglement-creating layer, nested between two parametrized rotation layers. The 4-qubit variational circuits of depth d=2 can be seen in Figure 1. For *q* qubits and depth *d*, the $R_{y}R_{z}$ variational circuit has 2q(d+1) parameters, single-qubit rotations, while the $R_{y}$ variational circuit has q(d+1) of them.

Our qubit Hamiltonians (Equation 3) and (Equation 5) contain terms with the Pauli *X* operators. To estimate their expectation values, we needed to measure them in a non-diagonal basis. As only computational-basis measurements are available on the quantum processors, we needed to utilize basis-switching single-qubit gates (post-rotations). For the Pauli operator *X*, we thus used a π/2 rotation around the *y* axis, performed by gate $R_{y}\left(\pi /2\right)$, and subsequently took measurements on a computational (*Z*-)basis.

Let us note that we do not need to estimate all of the Pauli terms in our Hamiltonians individually. We can save resources and reduce the number of required measurements and state preparations by grouping the Pauli operators that require the same post-rotations in the tensor product basis sets [27].

To access the real quantum devices, we used the publicly available cloud-based quantum computing platform IBM Quantum [49]. We found that access to real quantum processors is still limited for a larger systematic study of different types of errors and noise. Therefore, we heavily relied on classical simulations of quantum processors. The simulations were performed both in an ideal noise-free manner and including noise. Importantly, we used the noise-related parameters of the IBM quantum processors, which are publicly available (and changing) on a daily basis. Thereby, our simulations could more closely mimic the actual gates and computations executed on a real device.

The published noise-related data include an approximate noise model consisting of: (i) single-qubit and two-qubit gate errors—depolarizing errors followed by thermal-relaxation errors; (ii) single-qubit readout errors on all measurements; and (iii) all errors including gate, readout and thermal-relaxation errors. Since calibration data change frequently, in noisy simulation results we used the error model from the same calibration for most of the trials.

Finally, in some tests involving real quantum hardware, we employed a simple error-mitigation method. It uses least-squares fitting to obtain the error-mitigated outcome probabilities by using a calibration matrix [50].

## 3. Results and Discussion

There are several possible sources of error for the energies calculated by the VQE: first, the statistical errors in intermediate and final energy estimations, caused by the probabilistic nature of quantum mechanics; second, Hamiltonian representation and state-preparation ansatz errors, caused by approximations in the Hamiltonian (restricted basis set), as well as the space of states we search over, given by our parametrized quantum circuit; third, hardware errors present in noisy quantum devices running the quantum part of VQE. We study these using simulators of noisy devices, as well as real quantum processor runs. We will now investigate the influence of these errors, and discuss ways and the costs of mitigating them. We will do this for the H${}_{2}$ molecular Hamiltonians from the previous section.

Note that there are several factors which limit the number of gates/circuit repetitions we can execute. First, we have only limited access to the quantum processors, which severely restricts the number of shots we can use for each datapoint. Second, we do not have unlimited time either. Third, the gates themselves are noisy, so increasing system sizes or gate numbers will not work without serious error mitigation. Thus, simply increasing the number of repetitions/circuit size is not a solution to these errors, and we must work harder on error mitigation to obtain trustworthy results—molecular energies with chemical precision.

### 3.1. Number of Quantum Computer Calls

While searching for optimal parameters, the VQE optimization requires a quantum subroutine: prepare a candidate state and measure its energy. Even if we imagined having a noiseless quantum processor, its probabilistic nature and impossibility of measuring noncommuting observables simultaneously would force us to rely on averages over several measurements, which carry stochastic noise. We thus start by studying how the number of circuit (gate) evaluations on a quantum processor influences the convergence of the VQE algorithm, on ideal quantum devices.

Our goal is to understand and efficiently fight the influence of stochastic errors, inherent in VQE regardless of the quantum hardware. We want to optimally utilize a limited number of gate executions. For this, we have two options. First, we can increase the number of shots (controlled by the shots parameter in the programs) for each evaluation of readout probabilities. Second, we can increase the maximum number of iterations (the parameter maxite) allowed for the classical optimization subroutine.

Increasing the number of shots, improves the precision of the measurement outcomes for each of the terms in (Equation 5) or (Equation 3). This makes the energy function more stable and easier for the classical optimization routine to minimize. Meanwhile, increasing the allowed number of iterations for the classical optimizer provides it with a better chance to get out of existing local minima and/or to fine-tune the final output value, but again this comes at the cost of increasing the number of times a quantum computer needs to be accessed. Intuitively, it can be expected that increasing the number of quantum gate executions in either case increases the precision of the output. However, presently, the number of gate executions on a quantum computer is a limiting factor for most users—access to real quantum processors is either restricted or quite costly [51]. Additionally, an excessive number of quantum gate executions significantly impacts the real running time of the algorithm.

In this light, it would be interesting to find the minimum number of quantum gate executions that result in VQE output within chemical precision of the real energy value. In our study, we ran the algorithm for different settings of shots and maxiter parameters, on the two-qubit hydrogen molecule Hamiltonian, using the noiseless quantum simulator. Each combination of the settings was run 1000 times. We present the experimental results in detail in Table A2 (see Appendix B), and visualize it here in Figure 2 and Figure 3.

We visualized the data using a boxplot. The box shows the middle two quartiles of the data (interquartile range, IQR), with a marked median. Outliers are determined using Tukey fences [52]: a distance of 1.5 times the IQR in each direction, as in Figure 2a, or the extremes of our data, as in Figure 2b, where all of our computed energies fall above the red line, close to the box.

From Figure 2a, we can see that with an ideal quantum computer and unrestricted iterations of the SPSA algorithm, VQE performs increasingly better with the increase in the shots parameter. The limit of infinitely many shots is simulated using a statevector simulator, which calculates the outcome probabilities directly from the laws of quantum mechanics. It is important to note that when using a finite number of shots, the algorithm can also output energy values lower than the actual minimum energy. Thanks to stochastic noise, the estimates of outcome probabilities for the Pauli terms in (Equation 5), calculated from a limited number of shots, can be non-physical. To obtain realistic final optimized energies, one should thus invest in a precise final energy readout for the optimized state, using a larger number of shots.

This also raises another interesting question: can the inaccuracy in outcome probabilities be overcome by the SPSA algorithm? In other words, are the states discovered by SPSA actually close to the minimum eigenvector of the Hamiltonian, and is most of the energy spread seen in the results caused by the small number of shots? We recalculated the energies of the discovered states in Figure 2a, using the statevector simulator. The results (see Figure 2b) somewhat surprisingly show that indeed, most of the discovered states have real energies within the chemical accuracy.

In the context of trying to minimize quantum computer calls, this suggests that the VQE algorithm can be run with a small number of shots during the run of the classical optimization function and after the optimization produces the final result, the energy of the candidate state with minimum energy needs to be recalculated once more, using a much larger number of shots. However, for this to work, we have to use a classical optimizer which can deal well with noisy data. We could use Bayesian methods [53,54], which could be quite slow, with the need to invert large matrices to guess the next points for a search. Alternatively, we could use other optimization methods such as NEWUOA  [55], which can be adapted to work on noisy data. Note that, in the initial stages of this work, we have used the COBYLA optimization method [56], and concluded that very large numbers of shots were necessary for the method to converge at all. As described in the Section 2, in the end we decided on the use of SPSA, thanks to its natural noise resilience, as it explores random points around the current parameter vectors.

On the other hand, at least for our selected Hamiltonian, increasing the maxiter parameter and waiting longer for convergence seems to have diminishing returns after surpassing a value of approximately maxiter=100. We confirm this in detail in Figure 3. There, we simulate 1000 VQE calculations for various combinations of maxiter/shots settings. For the same shots parameter value, increasing the maxiter parameter only decreases the number of outliers, but the spread of the data remains almost unchanged. Both of these observations suggest that only a small fraction of runs can benefit from the increase in the maxiter parameter beyond 100.

To evaluate the energies more precisely after the last step, we recalculated energies using state vector simulator for shots parameter set to 512 and 1024, with maxiter settings 50,75 and 100. Results of this calculation can be found in Figure 4 and they suggest that the combination of shots = 1024 and maxiter = 75 already provides the median energy in the chemical accuracy for the 2-qubit H${}_{2}$ Hamiltonian.

### 3.2. Choice of Hamiltonian and State-Preparation Ansatz

In this section, we study the influence of the size and form of the quantum circuit used in the VQE calculation. The most straightforward way to reduce the complexity of the VQE algorithm is to find the smallest possible qubit representation of the studied Hamiltonian. Indeed, much effort today is focused on efficient encodings of fermionic Hamiltonians (see e.g., [57,58] and references therein), as getting rid of extra qubits (and operations on them) decreases noise and can avoid space limitations. To highlight the importance of finding minimal representations, we have chosen two equivalent variants of H${}_{2}$ molecule Hamiltonian, one using 2 and another one using 4 qubits (see Section 2).

Once we have chosen a Hamiltonian and its qubit implementation, VQE needs a choice of the variational circuit for state-preparation. We have a vast choice of parametrized gates, arrayed in multiple rounds. This results in varying numbers of classical parameters to be optimized, as well as different reachable quantum states (ansatz quality), with varying amounts of entanglement [59,60,61]. Here, we focus on two types of variational circuits—the $R_{y}$ and the $R_{y}R_{z}$ linear forms with a varying number of rounds, shown in Figure 1 for 4 qubits (the 2-qubit circuits are obvious simplifications).

Note that all the results in this subsection were obtained using a noiseless quantum simulation to highlight the influence of the Hamiltonian/ansatz choice in the ideal case. Of course, once we consider noise and statistical readout errors, the Hamiltonian/circuit choice translates to more errors, thanks to wider circuits with a larger number of gates.

In Figure 5, we depict our results for the convergence of the VQE algorithm, depending on the Hamiltonian choice (2- or 4-qubit), and the ansatz complexity (type and number of rounds).

The two-qubit Hamiltonian (Equation 5) turns out to be simple enough that the depth of the variational ansatz (number of layers) does not significantly influence the final optimized energies. Moreover, we can see that the simpler $R_{y}$ ansatz outperforms $R_{y}R_{z}$ ansatz. This confirms the intuition that the simplest ansatz containing the solution will also have the best performance, thanks to solution space having less parameters to optimize over. Additionally in noisy regime, simpler ansatz requires less gates for implementation, leading to further advantage.

On the other hand, for the 4-qubit Hamiltonian (Equation 3), both $R_{y}$ and $R_{y}R_{z}$ variational forms with 1 layer are not expressive enough and the search space does not contain the state with the lowest energy. Using two layers already alleviates this problem. Again, we see that the $R_{y}$ form performs better. Using a 12-parameter, two-level $R_{y}$ ansatz achieves better convergence than the 24-parameter, two-level $R_{y}R_{z}$ ansatz. Moreover, if we allow the $R_{y}$ form with 24 parameters as well, the circuit has depth 5 and outperforms the $R_{y}R_{z}$ ansatz even more significantly. This can be partially explained by the fact that for the same number of search space parameters, the $R_{y}$ ansatz can achieve more intricately entangled states, due to the increased number of entanglement layers.

These results only underline the importance of an efficient choice of Hamiltonian encoding and ansatz parametrization. More qubits mean the need for more intricate ansatzes, while introducing more statistical errors when estimating the energies of individual terms, and thus affecting the overall readout precision, even in an ideal (noiseless) case. Further, more complex Hamiltonians can cause the optimization algorithm to converge to a local minimum, thus causing a systematic error. This is in more detail treated in Appendix A.

### 3.3. Imperfect Quantum Devices

So far, we studied the convergence of the VQE algorithm assuming a perfect quantum computer available for the quantum subroutines. This analysis was possible thanks to relative simplicity of classically simulating the few-qubit quantum circuits used during our VQE algorithm runs. We wish to eventually scale up the calculations, and obtain results from the hybrid classical-quantum VQE which are not available classically. For this, we will have to rely on real quantum devices. Of course, current quantum processors are far from perfect, so the calculations will be inherently noisy, influencing also the convergence of VQE algorithm.

Let us group the possible errors into two basic types—gate errors and readout errors. This is a natural divide, since the number of measurement (readout) errors scales only with the number of qubits and can be mitigated with classical techniques. Meanwhile, the gate errors inevitably scale with both depth of the computational ansatz and the number of qubits, unless we use error correction/mitigation techniques.

In Figure 6 and Figure 7, we show the convergence of the VQE using simulations of noisy quantum processors. To obtain realistic parameters for the simulation, we used the error characterization of the real IBM devices, obtained directly from *qiskit*.

Our first observation is that the quality of calibration of recent experimental quantum processors changes frequently. Not only that, it significantly influences the convergence, as we demonstrate in Figure 6. There, we visualize the convergence of 1000 trials with the same settings, using calibration data from different dates, showing results differing by 10%.

Second, we observe that for these small circuits, the effect of measurement error is more significant than the gate errors. As we study this further in Figure 7 for the 4-qubit Hamiltonian (Equation 3), we find that readout errors still affect the convergence more, but the effect is less pronounced. In Figure 7a we use the 2-qubit, 1-round $R_{y}$ variational form, which requires 4 single-qubit gates + 1 CNOT, and up to 2 Pauli operators for readout. For Figure 7b, we recall results presented in Figure 5, and choose the 4-qubit, 2-round $R_{y}$ variational form, which requires 12 single-qubit gates + 6 CNOTs, and up to 2 Pauli operators for readout. The number of gates (circuit width × depth), and thus gate errors, grows faster than the number of estimation terms, which still scales with the circuit width. Thus, we expect that the effect of readout error will become less prominent for larger Hamiltonians compared to the gate errors. Moreover, we can also implement various readout error mitigating strategies—e.g., avoiding systematic bias by using bit flips before the final measurements.

### 3.4. Convergence of VQE on Real Quantum Hardware

Finally, in this subsection, we look at how the convergence predicted by the noisy simulator corresponds to convergence on real quantum hardware. For this, we performed an experimental runs on the *ibm_lagos* quantum processor, with maxiter = 75 and shots = 1024, and compared it to noisy simulation results with the same parameter settings. To mitigate the readout errors we performed a simple mitigation routine described at the beginning of this Section. In Figure 8, we observe that in fact the predictions agree fairly well with practice. Further, we tested our hypothesis that the discovered states are very close to optimal ones, but their energies have to be evaluated more precisely. To this end we calculated their energies using 40,000 shots on the *ibm_lagos* quantum processor (40,000 shots result in ≈0.5% error in energy calculations) as well as using a statevector simulator. In Figure 8, we can see that both of these techniques decrease the spread of the data points. However, only the statevector simulator consistently evaluates energies within the chemical accuracy from the exact ground-state value. This only shows that evaluating the energies using a large number of shots still suffers from hardware errors, which are not present when using the statevector simulator. We conclude that a simple VQE implementation on real quantum hardware does not *consistently* find the minimum energy within the chemical accuracy. Nevertheless, using both error-mitigation and precise energy readouts, one can obtain a trustworthy result, taking the lowest of the precisely evaluated energies.

## 4. Conclusions

In this work, we provided quantum chemistry calculations with the variational quantum eigensolver on real quantum hardware. We analyzed the influence of different types of errors on the convergence of VQE. We divided them into several categories and studied them separately to understand their impact and to find ways of avoiding them, while using our resources efficiently. For this, we used noiseless and noisy classical simulations, as well as publicly available superconducting quantum processors.

The errors resulting from the probabilistic nature of quantum mechanics significantly differ from the hardware errors. While the statistical error increases the spread of VQE outcomes around the optimum, gate and readout errors not only influence the spread of the outcomes, but often prevent the VQE from finding the optimum altogether.

If we could use noiseless hardware, the stochastic errors could be mitigated by increasing the number of shots, and obtaining precise energy estimates. This turns out not to be that important in the intermediate steps of iterative algorithm, but crucial at the end. Even with smaller numbers of shots per iteration, our optimization was able to find near-optimal states. However, to rule out unphysical results (energies below the actual minimum energy), and also to avoid overestimating the energy, we must invest shots into the final, precise readout. We learned that this approach can significantly reduce the overall cost (gate evaluations) for VQE, which is important, while access to real devices (and large runs) is limited, as well as time consuming.

Second, it is crucial to search for efficient encodings and good trial-state ansatzes (hardware, and problem-motivated). It saves on both gate evaluations, as well as on optimization complexity, reducing the number of required parameters to search over. We have demonstrated this in the comparison of the 4-qubit and 2-qubit Hamiltonians, and two types of ansatzes, with varying depths/numbers of parameters, for the same molecular hydrogen problem.

Finally, real devices’ parametrization changes with time, and it is crucial to choose to run your experiments at times when they are calibrated well. Moreover, the current levels of noise are prohibitive, and add up so that the final results do not straightforwardly reach the desired chemical precision. To avoid this, error mitigation must be used, for the readouts (which contribute a surprising amount of error) as well as inside of the circuit.

## Figures and Tables

**Figure 1 nanomaterials-12-00243-f001:**
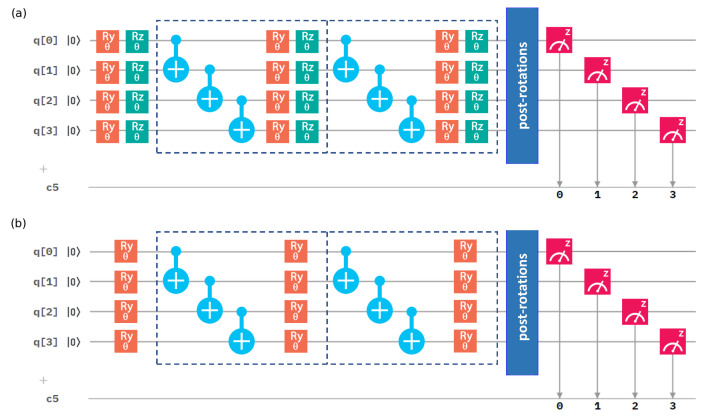
A scheme of the 4-qubit quantum circuits for calculations of the ground-state energy. The circuit depth was set to 2. Dashed lines represent blocks of two layers—the entangled layer and the rotation layer. (**a**) The $R_{y}R_{z}$ variational circuit. (**b**) The $R_{y}$ variational circuit.

**Figure 2 nanomaterials-12-00243-f002:**
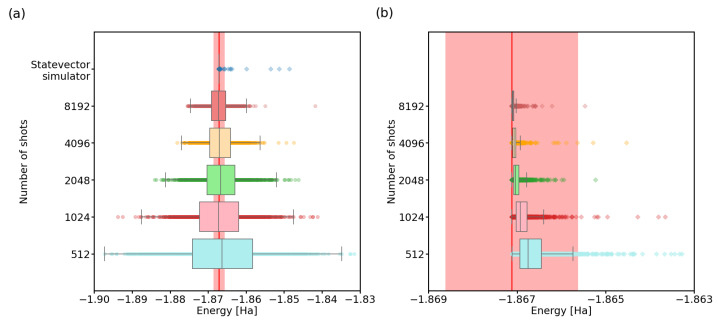
VQE energy estimate improvement with number of shots. We calculated the ground-state energy of H${}_{2}$ using the VQE on a two-qubit system, the SPSA optimization algorithm with unrestricted maximum interations, and the $R_{y}$ variational circuit accompanied by linear entanglement. The red line represents the physical ground-state energy (−1.86712 Hartree) and the light-red background represents the chemical accuracy regime (±0.0015 Hartree). (**a**) A boxplot of VQE results, with each box representing 1000 independent runs of the algorithm. (**b**) A statevector simulator calculation of the energies for the 1000 result states from (**a**). Note that all the energies are above the real ground-state energy.

**Figure 3 nanomaterials-12-00243-f003:**
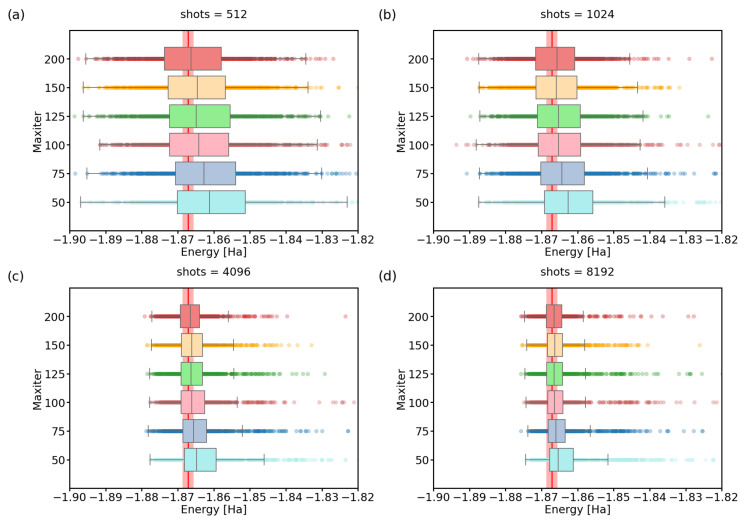
(**a**–**d**) A boxplot visualization of 1000 optimized ground-state energies for various maximum numbers of iterations of the SPSA optimizer. We performed calculations using the $R_{y}$ variational form. The red line represents the physical ground-state energy (−1.86712 Hartree) and the light-red background represents the chemical accuracy (±0.0015 Hartree). The  shots parameter was set to: (**a**) 512, (**b**) 1024, (**c**) 4096, (**d**) 8192.

**Figure 4 nanomaterials-12-00243-f004:**
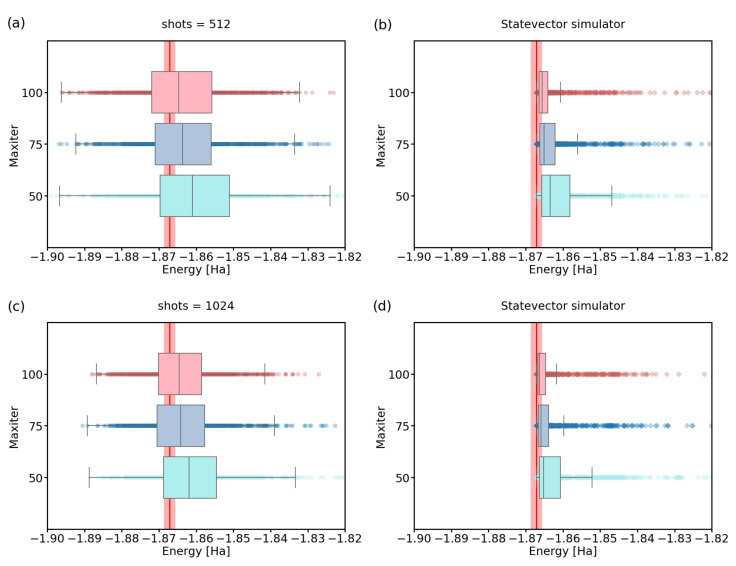
A recalculation of energies using statevector simulator for experiments with maxiter ∈{50,75,100}, and (**a**,**b**) shots = 512; (**c**,**d**) shots = 1024. The red line represents the physical ground-state energy (−1.86712 Hartree) and the light-red background represents the chemical accuracy (±0.0015 Hartree).

**Figure 5 nanomaterials-12-00243-f005:**
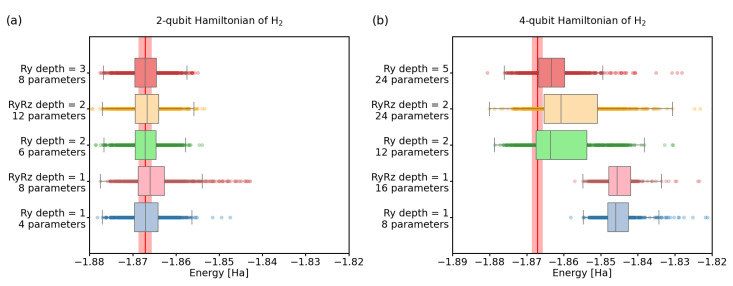
Comparison of the VQE energies for H${}_{2}$ using the $R_{y}$ and $R_{y}R_{z}$ variational circuit forms. We used a noiseless quantum simulator and 4096 shots; (**a**) 2-qubit Hamiltonian. (**b**) 4-qubit Hamiltonian. The red line represents the physical ground-state energy (−1.86712 Hartree) and the light-red background represents the chemical accuracy (±0.0015 Hartree).

**Figure 6 nanomaterials-12-00243-f006:**
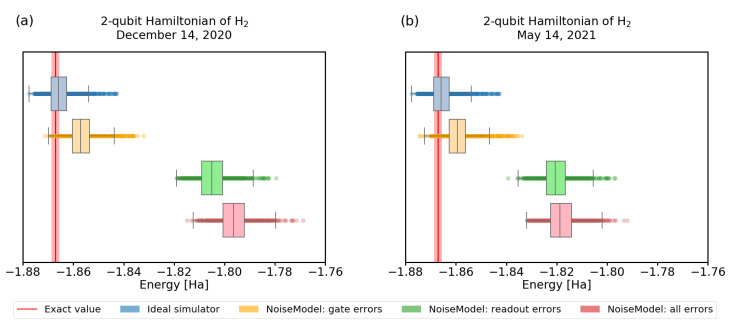
A comparison of VQE results with noise calibration from two different dates. We used quantum backend *ibmq_santiago*. All the other parameters in the simulation were kept the same, using 4096 shots and the $R_{y}R_{z}$ variational ansatz. The red line represents the physical ground-state energy (−1.86712 Hartree) and the light-red background represents the chemical accuracy (±0.0015 Hartree).

**Figure 7 nanomaterials-12-00243-f007:**
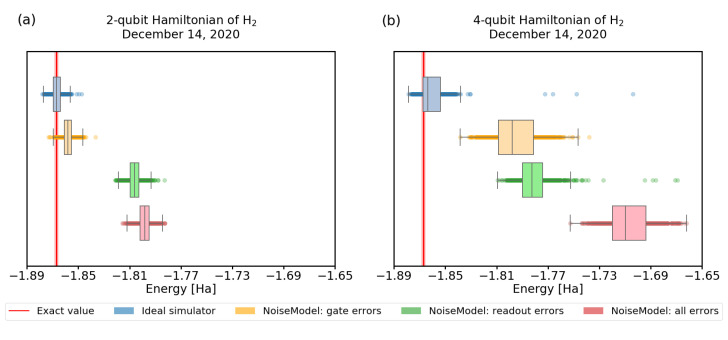
The calculations of H${}_{2}$ energy were performed using simulator of real quantum hardware using a 2 and a 4 qubit Hamiltonian. We used quantum processor *ibmq_santiago* with noise model from December 14 2020. The number of shots was set to 4096. We minimized energy with the SPSA optimizer: (**a**) in case of 2-qubit Hamiltonian we used unrestricted maximum number of iterations and the $R_{y}$ variational form with depth 1, (**b**) in case of 4-qubit Hamiltonian the maximum number of iterations was set to 400 and we used the $R_{y}$ form with depth 2. The red line represents the physical ground-state energy (−1.86712 Hartree) and the light-red background represents the chemical accuracy (±0.0015 Hartree).

**Figure 8 nanomaterials-12-00243-f008:**
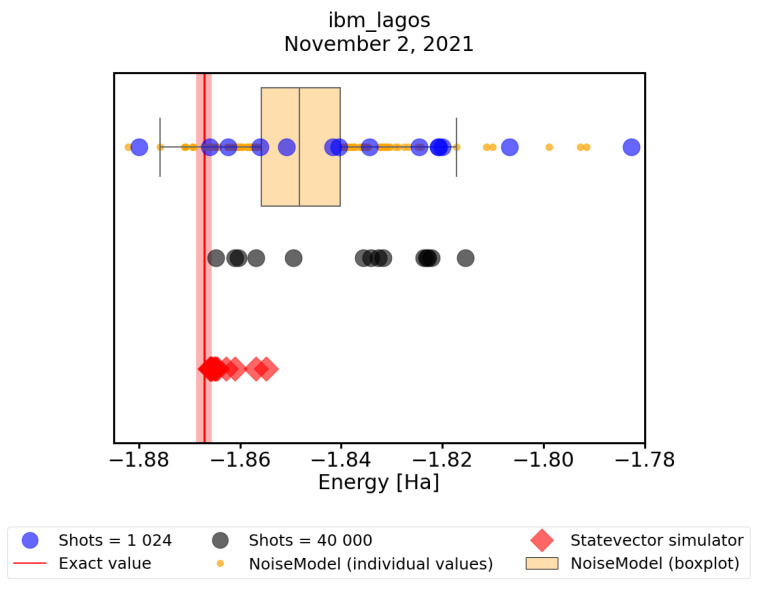
A comparison of results obtained with a noisy quantum simulator with error mitigation (orange points and box plot), with real runs of the *ibm_lagos* quantum processor (blue points). The energies of real runs were subsequently recalculated with greater precission using 40,000 shots of the *ibm_lagos* quantum processor (grey points). Finally, we recalculate their energies also with a statevector simulator (red diamonds). The red line represents the physical ground-state energy (−1.86712 Hartree) and the light-red background represents the chemical accuracy (±0.0015 Hartree).

## Data Availability

The data that support the findings of this study are available from the corresponding author upon request.

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
