# Peer review of "The Cost of Improving the Precision of the Variational Quantum Eigensolver for Quantum Chemistry"

_nanomaterials, 2022, doi:10.3390/nano12020243_

Round 1

Reviewer 1 Report

The manuscript fits into the up-to-date trend of the computational chemistry that exploits the idea of the use of quantum computing for quantum chemical calculations. This idea is rather original but accompanied with numerous obstacles. One of them deals with the stability and the solution process and this work tries overcoming it. For this purpose, the authors operate with Variational Quantum Eigensolver (an algorithm for calculating the physical Hamiltonian) and some errors influencing it (the artificial errors are created to study the correlation). Further, they calculate the eigenvalues of wave function of dihydrogen molecule using the simplest Slater-type basis set. The main result of this manuscript, for me, is that some level of noise upon intermediate calculations is required for successful calculation in total.

However, I have some suggestions relating to the manuscript.

  • Could the authors discuss the applicability of their approach to the calculations with larger basis sets, for example, very popular 6-31G(d)? Even if they do not perform such calculations, the working range of the approach is worth discussing.
  • The reference numbers are omitted everywhere in the manuscript (there are question marks instead them). Please, correct.
  • There are several points in the text with substandard English. For example (the list is not exhaustive):

Abstract: ‘an algorithm to calculate the minimum energy’ must be ‘an algorithm for calculating the minimum energy’

Abstract: ‘to obtain trustworthy calculations of molecular energies’ must be ‘to perform trustworthy calculations of molecular energies’

Line 77: ‘random fluctuations of parameters that are coupled to our qubits’ should be ‘‘random fluctuations of parameters coupled to our qubits’

Line 83: ‘Of course, our hope is fault-tolerant quantum processing, which is currently out of reach, thanks to precision and overhead requirements’ must be ‘Of course, our aim is fault-tolerant quantum processing, which is currently out of reach due to the precision and overhead requirements’

In several lines: Please, replace ‘in order to’ with ‘to’.

Lines 425-427: ‘In order to avoid this, error mitigation must be used for the readouts (which contribute a surprising amount of error) as well as inside of the circuit’ must be ‘To avoid this, error mitigation must be used both for the readouts (which contribute a surprising amount of error) and inside the circuit’.

These minor points are easily correctable and do not affect high level of the study. The manuscript is recommended for the publication.

Reviewer 2 Report

The paper focuses on the precision of VQE for quantum chemistry when using quantum devices. The paper is well written, and the results are sound. Unfortunately, the document misses all the references. Before publication, the authors should integrate that critical part of the paper.

Round 2

Reviewer 2 Report

The authors addressed all the reviewer's comments. The paper can be published as it is.